# Cold Start or Hot Start? Robust Slow Start in Congestion Control with *A Priori* Knowledge for Mobile Web Services

Paper #458, 8 pages body, 9 pages total

## Abstract

Mobile web services value a quick loading of contents in the first page, which is quantified by the above-the-fold time of the first page (first AFT) and is likely to fall into the slow start phase in congestion control. However, the widely deployed slow start mechanism is "cold start", which manually hardcodes the parameters and is not suitable for the first AFT of heterogeneous mobile web services. We revisit the slow start mechanism and find that it could be optimized with a priori knowledge. However, blindly relying on a priori knowledge is not robust enough to handle the fluctuating mobile networks and unpredictable application traffic. In this paper, we propose WiseStart, a "hot-start-based" slow start mechanism. WiseStart utilizes the priori knowledge to set the initial parameters, continuously probes the new connection to handle the fluctuating network conditions, and carefully adapts to the application-limit scenarios. We implement WiseStart in a popular mobile web service online in production. Comprehensive experiments demonstrate that WiseStart reduces the First AFT by 16.15% and the average RCT at connection establishment by 25.43% compared to the default slow start mechanism and other state-of-the-art baselines.

## Keywords

Transport Layer, Slow Start, Mobile Web Service, Hot Start

## 1 Introduction

Recently, a sharp increase in the usage of mobile web services has been observed. The latest statistics demonstrate that mobile users become the largest proportion of global Internet users [28]. Mobile web services value the content in the first page [7] – the loading time of contents above the fold (above-the-fold time, AFT)[1] is important. During the beginning of a connection, the connection has no information about the network, therefore existing transport protocols (e.g., TCP or QUIC) and congestion control algorithms has to probe the network. Such a process is called slow start, which is also responsible for the loading of those above-the-fold contents. However, the current design of slow start is usually a "cold start", where the start-up phase only relies on the hard-coded specifications rather than any up-to-date information about users or applications. This imposes great

---

[1]AFT is proposed by Google and defined as the time that the visible part of the page is loaded.

challenges on optimizing the performance of First AFT. Even if the hard-coded value is evolving with time (e.g., the initial congestion window (CWND) goes up from 4 [22] 20 years ago to 32 [23] in industry practices now), the static setting of slow start parameters does not match the need of optimizing First AFT under fluctuating and wide-ranging mobile network conditions.

Our large-scale measurement of one popular mobile web service further demonstrates the significance of optimizing slow start in mobile web services. A large amount of application data is concentrated when users start to access the application. We found that 19.38% of total requests, most of which contribute to the First AFT, are in the slow start phase. While, the "cold-start-based" slow start mechanism is insufficient for bandwidth utilization (§2.1). Our further parameter sweeping in §4.3 shows that even fine-tuning the parameter, as long as it's static across connections and time, is still suboptimal to the First AFT.

This motivates us to revisit the slow start mechanism and propose WiseStart, a new slow start mechanism for mobile web services. Our intuition is that *with a priori knowledge, we can directly use the appropriate initial sending rate*. In this case, slow start no longer suffers from blindly probing like cold start, but is able to utilize the previous application and network conditions and perform a "hot start". With the current routing policy, connections with the same pair of source IP and destination IP mainly experience the same path condition. Our large-scale measurements on the server side also demonstrate that the connection states are similar for the same access IP address. Therefore, WiseStart utilize the prior knowledge to set the initial sending parameters (§2.2).

However, merely setting the slow start parameters based on a priori knowledge can be fatal in some cases. WiseStart should be robust to the fluctuations of both mobile network conditions and application traffic. First, the available bandwidth can fluctuate and be different from the prior knowledge, especially in mobile networks [17]. Blindly reusing historical information may lead to performance degradation when bandwidth fluctuates so much that the prior knowledge is invalid. In response, WiseStart continuously probes the available bandwidth to update the prediction for the new connection. WiseStart estimates the available capacity based on the initial few ACK packets and decides whether to continue up-probing or drain the queue. Second, the application traffic pattern affects the estimation of the available capacity as well. The request initiation is affected by user

behavior, which is unpredictable and might conflict with the connection's state. When the connection is intermittently in the application-limit state [11] and a request is initiated, the measurement of capacity may be inaccurate. This in turn affects the effectiveness of the slow start mechanism. Therefore, we design an application-limit detection mechanism in WiseStart and adapt WiseStart to cater for the connection states and traffic patterns (§2.3, §3).

We implement WiseStart atop Cubic based on QUIC in the production environment of an M Company's popular mobile web service with O(10M) daily active users. We compare WiseStart with the default slow start mechanism and other state-of-the-art baselines. Comprehensive experiments demonstrate that WiseStart achieves consistent high performance, reduces the First AFT by 16.15%, and reduces the average RCT at the beginning of the connection by 25.43% against baselines (§4).

In summary, our key contributions in this paper are:
- Through online measurement of a mobile web service in a production environment, we expose the problems with the First AFT of mobile web services (§2).
- We propose WiseStart, an adaptive slow start mechanism, which reuses historical connection information and adapts to the fluctuating mobile networks and the irregular impulse requests of mobile applications (§3).
- We deploy and evaluate WiseStart in a mobile web service at M Company. Our extensive experiments show that WiseStart achieves consistent high performance both in production environments and on various locally emulated network conditions (§4).

## 2 Motivation

In this section, we conduct a measurement study on the First AFT of a popular mobile web service. We first explain that the bad performance of first AFR comes from the mismatch between application requirements and transport layer capabilities in the slow start phase (§2.1). Then, we present the opportunity of our design choices to address the mismatch (§2.2), and describe the design challenges (§2.3).

### 2.1 Why optimize the slow start?

**Significant First AFT performance.** For mobile web services, the performance of First AFT is critical. If the application takes too long to load, users may give up waiting and switch to other applications [24]. Among all the contents on a page, the content of the first screen (i.e., above the fold) is undoubtedly the most important – users will be partially satisfied if contents in their sights (above the fold, technically) are ready. Google reports that ads appearing above the fold have 30% higher visibility than others [5]. Moreover, such a phenomenon happens quite often on mobile phones. For example, mobile users sometimes switch between multiple applications. The operating system of mobile phones will

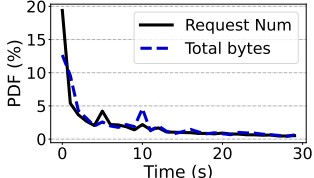
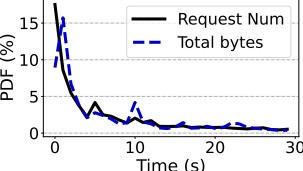

(a) Application Initialization  (b) Connection Establishment

**Figure 1: Distribution of the number of requests and total bytes transferred within each second during the use of the application. (a) Probability density function (PDF) within 30s after application initialization. (b) Probability density function (PDF) within 30s after connection establishment.**

periodically clean up the applications running in the background (e.g., minutes in Android [2]). In this case, if a user stays several minutes at another application and switches back, the connection has to be set up again and the content has to be reloaded [1].

We make an in-depth measurement study on one region of a popular mobile web service of M Company with O(10M) daily active users, containing one million requests over 10 days. We count the number of requests and total bytes transferred within each second in each session and present the probability density within 30 seconds after application initialization and connection establishment respectively. As shown in Fig. 1, 19.38% of the requests were initiated within 1 second after application initialization, and their overall data volume accounted for 12.67% of the full connection data. This indicates that even one user might use the application for a long time, and a considerable amount of requests are only from the first second. A similar situation is observed after the connection establishment, and the frequency of new connections is 5.72 times per user per day. Therefore, improving the load speed of the first page content is crucial to the user experience.

**Mismatch in the slow start phase.** At the beginning of the connection establishment, the transport layer starts from a small CWND or sending rate and gradually increases them to probe the available capacity. This process is the slow start mechanism. However, during the slow start phase, it is not possible to utilize the bandwidth well. Taking the 2Mbps bandwidth and 200ms RTT as an example, using the default slow start algorithm, it takes five RTTs to reach the available bandwidth, which will last one second. As the capacity cannot be efficiently used, the request completion time (RCT) and First AFT increases, and the user experience deteriorates. To investigate the performance difference of the first second of the overall connection, we also measure the RCT of requests sent within 1s after connection establishment and the RCT of the overall connection. Results show that the RCT for the first second is 1.5 times of the average of the overall connection, which further exacerbates the first AFT.

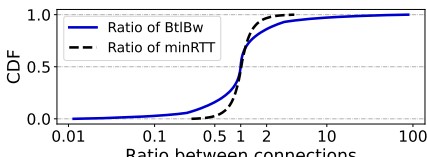

Figure 2: Distributions of the ratio of BtlBw and minRTT between two sequential connections from the same peer IP address, in our large-scale passive measurement on the server side.

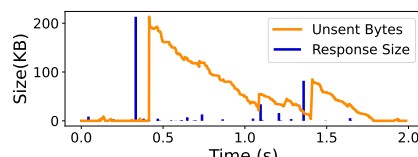

Figure 3: An example of the application-limit state. The blue lines show the response size of the request sent at that time, and the orange line shows the unsent bytes on the server side.

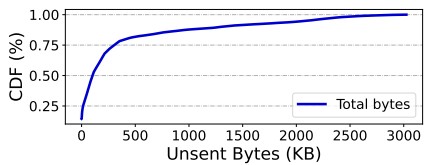

Figure 4: Distribution of the unsent bytes on the server side. There are 14.46% cases that there is no application data waiting to be sent at that time.

## 2.2 Design Opportunity

Now that a static setting of initial CWND is suboptimal, our intuition is that we can directly use the a priori knowledge to set the initial CWND and sending rate. Fortunately, for content providers, it is indeed possible to obtain relevant information from historical connections or parallel connections from the same user. On the one hand, mobile web services may establish multiple connections in a short period when users switch between foreground and background. Two applications of Company M have similar scenarios, with 1.84 and 3.52 connections per user per day, respectively. On the other hand, the same access IP address belongs to the same AP or base station, so the connections with the same access IP have similar access network conditions. Moreover, connections with the same pair of source IP and destination IP mainly experience the same path. Therefore, for content providers, connections with the same client IP will experience similar path conditions.

Our measurements further verify the above observation on the similarity of the performance between multiple connections from the same IP. We record the minimum RTT (minRTT) and maximum available bandwidth (BtlBw) of each connection in another measurement campaign on our mobile web service. Our measurement lasts 14 days, and contains information of 2.3 million IPs with 11.65 million connections. We calculate the ratio of the minRTT (and BtlBw) between two connections from the same client IP, and plot the Cumulative Distribution Function (CDF) of the ratio results in Fig. 2. Note that the two connections are not necessary to be simultaneous – we calculate the ratio for all connections sharing the same client IP in the 14-day measurement. We find that in more than 60% of the cases, the difference ratio of the minRTT of two connections from one client IP is less than 20%. And from the view of absolute RTT values, in 70% of the cases, the difference is less than 6 ms. The ratio of BtlBw shows consistent similarity as well.

## 2.3 Design Challenges

WiseStart improves the efficiency of the slow start with the a priori knowledge. However, blindly relying on a priori knowledge is not robust enough in the following two aspects:
**Robust to mobile network.** In mobile scenarios, the path

capacity may be variable. Mobile network conditions fluctuate due to wireless channel fading, user moving or competing flows [26, 28]. Reports show that fluctuations in bandwidth can reach more than two times even for one single connection [17], let alone reusing the information of connections from a different time. Just as our measurement in Fig. 2 shows, there are 8.5% cases where BtlBw changes by more than 5 times. When fluctuations are so large that a priori knowledge is invalid, blindly setting the initial CWND based on previous (probably large) CWNDs may result in a large number of packet losses due to overshooting the degraded network, or unused available capacity due to an unnecessary small initial CWND. Therefore, WiseStart needs to probe the available bandwidth as well for new connections.

**Robust to application traffic.** Meanwhile, the application traffic, determined by unpredictable user behaviors, is also fluctuating in mobile web services. Thus, when the application plans to send a request, if the connection is in the application-limit state, the previous parameters will be less effective as well. Fig. 3 shows an example: if the user's click behavior is sparse, there may not be enough application data to fill the capacity. We replay real-world application traces extracted from M Company and find that there are 14.46% of the time that there is no application data, as shown in Fig. 4. In this case, the estimated bandwidth is limited by the rate the application generates data and can not reflect the bottleneck bandwidth. The inaccurate bandwidth measurement affects the capacity estimation of new connections, which in turn affects the timing of exiting WiseStart. Therefore, WiseStart also needs to adapt the design to handle different, mainly application-limit, states of the connection.

## 3 Design

In this section, we present the design of WiseStart. We first give an overview (§3.1) and then describe three key design details: (1) How to store and reuse the prior knowledge (§3.2). (2) How to quickly and accurately probe new connections (§3.3). (3) How to adapt the bandwidth estimation methods to the application-limit scenario (§3.4).

### 3.1 Design Overview

As shown in Fig. 5, WiseStart has three key design points:

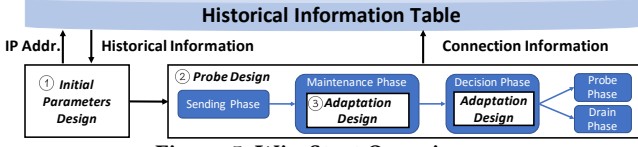

**Figure 5: WiseStart Overview.**

**Initial parameters.** First, WiseStart records the bandwidth and minRTT of each connection. When a new connection is established, it looks up the historical record based on the IP address and sets the initial CWND and sending rate.

**Probe.** Second, WiseStart continuously probes the new connection through three phases. By estimating the bandwidth and minRTT of the new connection, WiseStart decides whether to continue probing or to drain over-sent packets.

**Adaptation.** Third, we adapt the whole design for the possible application-limit scenarios.

Wisestart utilizes CWND to manage the in-flight bytes and enables pacing to regulate the sending rate. It can be integrated with any congestion control algorithm that includes a slow start phase. Wisestart is implemented on the sender side and does not require receiver-side modifications. Algorithm 1 describes the pseudo-code of how WiseStart works specifically.

### 3.2 Set initial parameters with priori knowledge

One key design choice of WiseStart is to store the historical connection information and set the initial CWND or sending rate of the new connections accordingly. Ideally, the CWND should equal to the path capacity. Therefore, WiseStart stores the round-trip propagation time (RTProp) and the bottleneck bandwidth (BtlBw) to calculate the capacity.

**Record and store historical information.** WiseStart uses a LRU hash table on the server side to cache information of historical connections. Each table entry comprises four fields: record timestamp, peer IP address, the maximum delivery rate (BtlBW), and the minimum RTT (RTProp). When a connection is closed, WiseStart records the above four states.

**Set initial parameters.** When a new connection is established, WiseStart looks up the hash table for BtlBw (labeled as Last_BtlBw) and RTProp (labeled as Last_RTProp) based on the peer IP. WiseStart sets the initial CWND as $CWND = Last\_BtlBw * Last\_RTProp$, and paces the packets in the initial CWND to avoid packet loss caused by this large burst. Considering the requirement of probing the new connection, WiseStart sets the pacing rate as $Pacing\_rate = 2 * Last\_BtlBw$. If there is no record in the hash table for that IP address, WiseStart falls back to the default slow start mechanism which increases CWND from 10 MSS.

### 3.3 Probe the new connection

Considering the fluctuation of mobile networks, it is not enough to set the initial parameters, but also to continue probing the path. WiseStart estimates the bandwidth of the

---

**Algorithm 1:** WiseStart Algorithm

**1 Initialization:**
**2**    $Last\_RTProp, Last\_BtlBw \leftarrow LookUp\ (IP)$;
**3**    $\gamma \leftarrow 6, \kappa \leftarrow 20 * \text{MSS}$;
**4 On connection establishment** (§3.2):
**5 begin**
**6**    $CWND \leftarrow Last\_BtlBw * Last\_RTProp$;
**7**    $Pacing\_Rate \leftarrow 2 * Last\_BtlBw$;
**8 end**
**9 On each packet sent before the 1st ACK** (§3.3.1):
**10 begin**
**11**    **if** $Inflight \geq min(CWND, \kappa)$ **then**
**12**       $Pacing\_Rate \leftarrow 0.5 * Last\_BtlBw$
**13**    **end**
**14**    **if** $Inflight \geq min(CWND, 1.5 * \kappa)$ **then**
**15**       $Pacing\_Rate \leftarrow 1 * Last\_BtlBw$
**16**    **end**
**17 end**
**18 On the 1st - $\gamma$ − 1th ACK** (§3.3.2):
**19 begin**
**20**    $CWND \leftarrow Cur\_Inflight$;
**21**    $DisablePacing()$;
**22 end**
**23 On the $\gamma$th ACK** (§3.3.3):
**24 begin**
**25**    $Est\_BtlBw \leftarrow Calcu()$;      /* Eq. (1) */
**26**    **if** $CWND \geq Est\_BtlBw * Est\_RTProp$ **then**
**27**       $Drain()$;
**28**    **else**
**29**       $Probe()$;
**30**    **end**
**31 end**

---

new connection by measuring the delivery rate of first several packets and makes further decisions on whether to increase the CWND or drain the queue subsequently. WiseStart performs path probe through the following three phases.

**3.3.1 Sending Phase** The Sending phase is from setting the initial parameters to receiving the first ACK. In the Sending phase, WiseStart sends the first several packets at a higher rate (2*Last_BtlBw) to probe higher bandwidth, and sends subsequent packets at a lower send rate (0.5*Last_BtlBwd) to avoid packet loss due to the over-speed sending. $\kappa$ in Algorithm 1 should be higher than the bytes acknowledged by the first $\gamma$ ACK packets, which is the number of ACK packets used to estimate the available bandwidth.

**3.3.2 Maintenance Phase** The Maintenance phase is from receiving the first ACK packet to receiving the $\gamma$th ACK packet. In the Maintenance phase, WiseStart converges CWND

to the BDP and continuously records the information from the received ACKs. First, when the first ACK is received, WiseStart disables the pacing mechanism and sets CWND as the amount of inflight data subtracts the number of bytes of lost packets. Therefore, packets are allowed to send only when ACK packets are received, and the CWND gradually converges to the path BDP. Besides, WiseStart records the information from ACK packets, including the receiving time, the number of acknowledged bytes, and the estimated RTT.

### 3.3.3 Decision Phase

The Decision phase occurs when receiving the $\gamma$th ACK packet. WiseStart estimates the available bandwidth (labeled as Est_BtlBw) and RTProp (labeled as Est_RTProp) of the new connection and decides whether to probe or drain subsequently. First, WiseStart estimates Est_RTProp as the minRTT over the first $\gamma$ ACK packets, and computes Est_BtlBw based on the acknowledged bytes:

$$Est\_BtlBw = \frac{\sum_{i=2}^{\gamma} Ack\_Bytes_i}{Recv\_Tstmp_\gamma - Recv\_Tstmp_1} \quad (1)$$

Second, WiseStart calculates Est_BDP as $Est\_BDP = Est\_BtlBw * Est\_RTProp$ and compares it with CWND. If Est_BDP is larger than CWND, WiseStart enters the Probe phase; otherwise, WiseStart enters the Drain phase.

In the Probe phase, WiseStart increases the CWND from current CWND, following the default slow start algorithm of exponentially increasing CWND. In the Drain phase, WiseStart converges to the BDP according to the strategy of the Maintenance phase.

## 3.4 Adapt to application-limit scenarios

Implicit in the design of probing the new connection (§3.3) is the assumption that the amount of application data is sufficient to probe the connection bandwidth. The assumption exits in two phases:

- **Maintenance phase.** Maintaining CWND as the inflight bytes implicitly assumes that the inflight bytes reflect the BDP of the path. However, when application data is insufficient, inflight bytes only reflect the amount of data generated by the application. The CWND may be so small that it cannot be sent when the application has more data in the Maintenance phase.
- **Decision phase.** When estimating Est_BtlBw, there is an assumption that the bottleneck is in the network. However, if the application data is insufficient, the Est_BtlBw is limited by the rate that application generates data and can not reflect the bottleneck bandwidth of the path.

However, as §2.3 stated, transient application data shortage occurs in mobile web services. Therefore, WiseStart adapts to application-limit scenarios for the above two issues.

**Detection of application-limit state.** WiseStart performs real-time detection of application-limit state. Previous solutions [11] are atop CWND-based algorithms, whose detection criterion is coarse-grained that whether the inflight bytes fill the current CWND. However, a fine-grained and rate-based-algorithm-supported detection is needed. WiseStart measures the actual sending rate (labeled as Est_SendRate) of the sender and compares it with the pacing rate set by the sender to decide if the current sending behavior is limited by the pacing rate or by the application. The measurement of Est_SendRate lasts in all three phases in §3.3.

**Adaption to the Maintenance phase and Decision phase.** In the Maintenance phase, when receiving the first ACK, WiseStart compares Est_SendRate with the setting rate. If it is in the application-limit state, WiseStart continues pacing the packets as the Sending phase. While in the Decision phase, when receiving the $\gamma$th ACK packet, WiseStart compares Est_BtlBw with Est_SendRate if it is in the application-limit state. If Est_BtlBw is lower than Est_SendRate, which means the actual sending data fills the path, WiseStart enters the Drain phase; if not, WiseStart enters the Probe phase.

## 4 Evaluation

We first introduce the implementation and experimental setup (§4.1). We then evaluate WiseStart as follows:

- **Performance in the real world.** We implement WiseStart in a popular mobile web service. Experiments with real users show that WiseStart reduces RCT within 1s of connection establishment by 16.15%, with acceptable computation and memory overhead (§4.2).
- **Consistent high performance.** WiseStart achieves great improvement under different network conditions, and reduces the First AFT by 25.43% (§4.3).
- **Design Effectiveness.** We analyze WiseStart's effectiveness of handling the fluctuating bandwidth and the application-limit state. We also investigate the fairness and friendliness of WiseStart (§4.4).

## 4.1 Experimental Setup

We implement WiseStart atop Cubic based on QUIC[2] in user space. WiseStart only requires modification on the sender side. We evaluate WiseStart in both large scale production environment (§4.2) and emulated networks (§4.3, §4.4).

**Large scale production environment.** We implement WiseStart in a popular mobile web service of M Company, with O(10M) daily active users. We manually modified the server-side settings to allow a fraction of users to use WiseStart as the slow start mechanism. We measure the performance for 73 hours and collect 86 million request logs from more than 50 countries and regions. When users access to the application, the client establishes a persistent connection with the frontend server and send requests. The load balancer

---

[2] We use an IETF QUIC implementation, ngtcp2 [3].

hashes the request to one of the front-end servers of the cluster based on the client IP address. Therefore, WiseStart is deployed on the front-end server and stores the connection information locally in the form of a static hash table. The connection of the same peer IP address will be always routed to the same front-end server.

**Emulated environment.** We also evaluate WiseStart in a controlled environment by emulating different network conditions with Mahimahi [20] and replaying real application traces. In our testbed evaluation, we implement a simple request-response messaging application atop WiseStart, which sends requests and responses with application traces, and collects statistics for evaluation.

**Baselines.** We compare the performance of WiseStart respectively with several baselines to demonstrate its effectiveness.

- *S-Cubic* is a newly proposed slow start approach which reuses the historical information and enters into congestion avoidance directly at the first ACK [12].
- *Cubic32* and *Cubic64* statically set the initial CWND to 32 and 64 respectively based on Cubic.
- *Cubic* and *BBR* are the default algorithms.

### 4.2 Real-world performance

**RCT Performance.** We collect RCT from client side and show the RCT within 1 second of the connection establishment in Fig. 6. WiseStart achieves the best performance in real-world scenarios. WiseStart is able to reduce the RCT within 1 second of connection establishment by 8.56% to 16.15% in real-world scenarios, with a reduction in tail RCT of 22.65% to 52.34%. Although S-Cubic utilizes historical information, it directly enters into the congestion avoidance phase without probing the new connection. On the one hand, the historical information may fail due to the fluctuation of mobile networks. On the other hand, S-Cubic does not perform application-limit adaptation, which also makes the CWND setting much lower when exiting slow start. Therefore, S-Cubic does not achieve good performance and WiseStart reduces the RCT by 16.15% compared to S-Cubic. Cubic64 show improvements compared to Cubic. However, due to the wide range of bandwidth under mobile networks, the fixed initial CWND may be suboptimal, which is insufficient at some times and is too large that causes packet loss at other times. Therefore, both Cubic32 and Cubic64 experience long tail latency, and WiseStart also reduces the RCT by 15.55% and 8.56% compared to Cubic32 and Cubic64, respectively. As for Cubic and BBR, the performance is poorer due to the small initial CWND and sending rate.

**Overhead.** We count the hit rate and occupied memory of hash table in hours. As shown in Fig. 7, the hit rate of the hash table is only 24.59% within one hour after WiseStart first deployed. The hit rate gradually increases as the num-

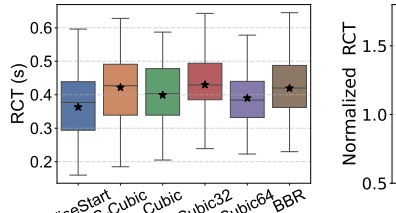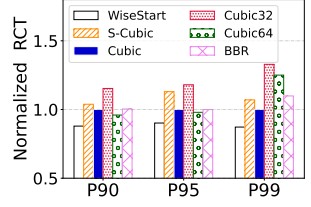

(a) Distribution of the RCTs.     (b) Normalized tail RCTs.

**Figure 6: In real world experiments, WiseStart brought 16.15% reduction on average RCT, and 52.34% for the $99^{th}$ percentile (the tail) completion time within one second of the connection establishment.**

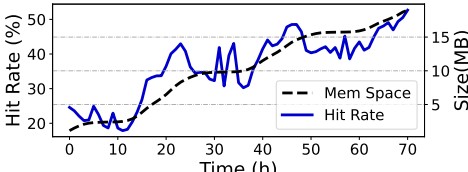

**Figure 7: The hit rate and memory usage of the hash table.**

| Parameter | Value Range (Min - Max) |
|---|---|
| RTT(ms) | 10 - 50, 50 - 100, 100 - 150, 150 - 300 |
| RTT Jitter / RTT | 0 - 0.2, Jitter max = 20ms |
| Loss rate(%) | 0 - 0, 0 - 0.1, 0.1 - 5 |
| Buffer / BDP | 0.3 - 0.9, 0.9 - 1.1, 1.1 - 1.5 |

**Table 1: Network condition parameters.**

ber of accessed users increases, and reaches 52.66% after 60 hours. This means that the about half of users can reuse historical connection information. For memory occupation, since we allocate memory dynamically, the memory occupation also increases with the number of accessed users, and the final memory occupation in steady state is about 19MB. The additional CPU utilization of WiseStart is 3.6%.

### 4.3 Improvement on First ATF

We evaluate the improvement of WiseStart on the First AFT through controlled experiments in emulated scenarios with real user traces. First, we collect and replay traces from real application, and mark the first-screen requests. Second, we use the network traces collected and used in previous works [8, 15, 16, 18, 19] to emulate real mobile network environment through Mahimahi [20]. Our emulated experiments involve three scenarios: stationary cellular scenario, highly variable scenario and WiFi scenario, with a total of 70 traces. We set other network parameters randomly selected within the range in Tab. 1. WiseStart and S-Cubic run through all scenarios sequentially as the other algorithms did, without pre-recording any information about the scenarios. This means that the historical information stored is about the previous scenario, which may be significantly different from the current scenario.

We record the final completion time of all first-screen requests and show the results of all scenarios in Fig. 8. Fig. 8a

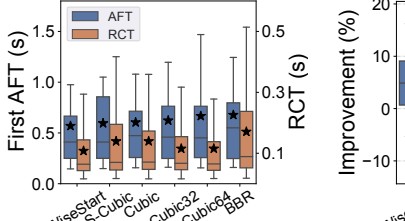
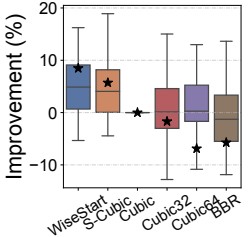

(a) First AFTs and RCTs of the first-screen requests.

(b) Improvement of First AFT compared to Cubic.

**Figure 8: WiseStart reduces the overall First AFT, and shows a consistent improvement in 91.43% of the scenarios.**

shows the Tukey boxplot of the First AFT and the RCT of the first-screen requests. To further analyze the improvement of WiseStart, we record the reduction in First AFT for different algorithms compared to Cubic under each scenario in Fig. 8b, i.e., positive values imply performance gains.

WiseStart achieves the lowest First AFT, and reduces the overall First AFT by a median of 5.84% to 25.43%, and 9.64% to 36.81% in the $95^{th}$ percentile. As shown in Fig. 8b, WiseStart shows a consistent improvement and reduces First AFT in 91.43% of the scenarios, with an average reduction of 8.5%. S-Cubic also reduces First AFT in 81.42% of the scenarios, with an average reduction of 5.7%. However, since S-Cubic does not probe new connections, when meeting large new bandwidth, it takes a long time to converge and a severe performance degradation occurs. This demonstrates the necessity of exploring new connections. Cubic32 and Cubic64 only have improvements in about 60% of the scenarios and the overall performance is degraded. Increasing the initial CWND could improve the link utilization to some extent, while it may also introduce significant packet loss. Cubic64 and Cubic32 have 21.67% additional packet loss compared to WiseStart, and in some scenarios the loss rate even reaches 1.56%. BBR performs worse than Cubic, and it is because BBR tends to overestimate RTProp and brings packet loss.

### 4.4 WiseStart Deep Dive

We analyze WiseStart's effectiveness of handling the fluctuating bandwidth (§4.4.1) and the application-limit state (§4.4.2). Then, we investigate its fairness (§4.4.3).

**4.4.1 Fluctuating bandwidth** We analyze the resilience of WiseStart to the fluctuating bandwidth of new connections. We set the base bandwidth as 24 Mbps with 40 ms RTT and buffer size of 1 BDP. We vary the ratio of bandwidth of the new connection to the base bandwidth from 0.2 to 2.0 and evaluate the performance on the new connection. We perform experiments using long flows (1024KB) and record the flow completion time (FCT) and loss bytes. For WiseStart and S-Cubic, we firstly run an experiment through base bandwidth once to record the information of the base connection in WiseStart. Then we run a new experiment through the new connection. In addition, we directly set the

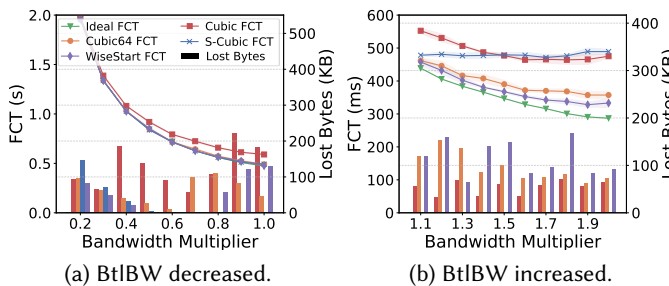

(a) BtlBW decreased.  (b) BtlBW increased.

**Figure 9: WiseStart presents robustness to fluctuating bandwidths. WiseStart reduces packet loss by about 20% when bandwidth decreases and reduces FCT by 29.79% when bandwidth increases.**

pacing rate and CWND as the bandwidth and BDP of the new connection respectively, as Ideal.

When the bandwidth of the new connection is lower than that of the old connection, there is not much space for optimization in the slow start mechanism due to the low bandwidth. As shown in Fig. 9a, WiseStart, S-Cubic and Cubic64 are all consistent with the Ideal for the FCT. WiseStart reduces packet loss because it probes the new connection and drains additional queues. In contrast, S-Cubic directly enters the congestion avoidance phase after reusing old connection information. Since the buffer is set as one BDP, S-Cubic drops a lot of packets when the new bandwidth is reduced to less than half of the old connection. For strategies of increasing the initial CWND (e.g. Cubic64), when the capacity of the new connection is smaller than the initial CWND, packet loss occurs at the initial busrt and it will enter the congestion avoidance phase immediately. In this case, Cubic suffers from more packet loss because it increases the CWND and will exit slow start after one RTT of the packet loss.

For the scenarios where the bandwidth of the new connection is increased compared to the old one, none of the mechanisms achieve the Ideal because the accurate information about the connection is not available at establishment. WiseStart is the closest to the Ideal, reducing the FCT by 29.79% compared to Cubic when the bandwidth is increased by a factor of two (Fig. 9b). Increasing the initial CWND (e.g. Cubic64) can also accelerate the slow start, while its improvement becomes worse as the difference between the path capacity and the initial CWND increases. In addition, WiseStart, Cubic and Cubic64 all suffer from packet loss. This is because that these algorithms all inevitably use loss to determine whether to exit the slow start mechanism. S-Cubic does not drop packets because it does not probe new connections. However, its performance degrades as the difference between the bandwidth of the new connection and the old one increases, and is sometimes even inferior to Cubic.

**4.4.2 Application-limit state** To evaluate the effectiveness of WiseStart's adaptation for the application-limit state, we additionally disable the adaptation module (WiseStart-

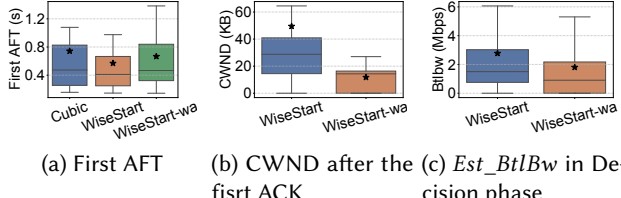

(a) First AFT    (b) CWND after the (c) $Est\_BtlBw$ in De-
                  fisrt ACK      cision phase

**Figure 10: The adaptation to application-limit state contributes 14.6% to the reduction of the First AFT.**

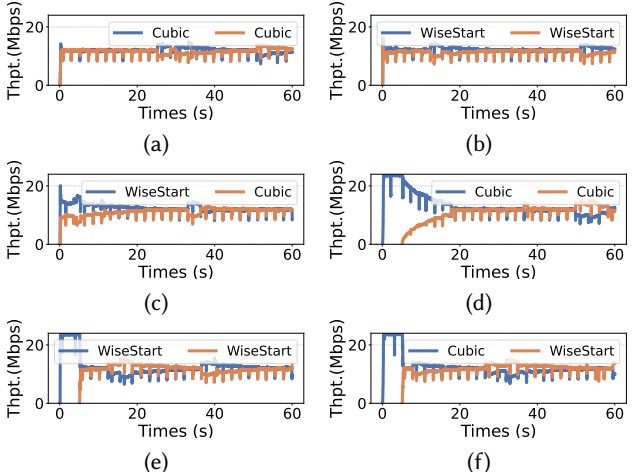

**Figure 11: Temporal dynamics of two competing flows. (a)(b)(c) Start simultaneously, (d)(e)(f) Start with 5s interval**

wa) and compare its performance with WiseStart. The experimental setup is the same as §4.3. As shown in Fig. 10a, WiseStart reduces the First AFT by 14.6% on average and the tail First AFT by 31.98% compared to WiseStart-wa. On the one hand, when the first ACK is received, WiseStart-wa sets the CWND to the inflight bytes, which might be smaller than the BDP. As shown in Fig. 10b, CWND after the first ACK of WiseStart-wa is 76.26% lower than WiseStart, which is 37.8 KB lower on average. On the other hand, the new connection BDP estimated by WiseStart-wa is lower (Fig. 10c), and thus WiseStart-wa may enter the Drain phase incorrectly. WiseStart-wa had a 4.5% higher probability of entering the Drain phase than WiseStart. Theses results demonstrate the necessity of the adaptation for application-limit states.

**4.4.3 Fairness and friendliness** We evaluate WiseStart's fairness and friendliness. We set up two server-client pairs sharing the same bottleneck link with 24 Mbps bandwidth, 40ms RTT, and buffer size of 1 BDP. We consider both the case where two flows start simultaneously and where the latter flow starts after the former one converges (5 seconds later in our experiment). Note that each WiseStart flow has run through the bottleneck alone in advance and stored the historical information. As shown in Fig. 11, WiseStart achieves a high degree of fairness towards its own competing flows. Also, when a WiseStart flow enters a link with a converged

WiseStart flow, WiseStart converges significantly faster than Cubic (about 10 seconds faster in Fig. 11e). As for the friendliness, the WiseStart flow can achieve the same throughput with Cubic flow, and converge fast when entering the path of existing Cubic flows.

## 5 Related work

In the last decades, bandwidth has grown so rapidly that the default TCP slow start mechanism can no longer accommodate current bandwidth conditions. When the default initial window within Linux was designed, the average connection speed was about 1.7 Mbps [10]. While the current report shows that as of late 2021, the median Wi-Fi bandwidth is 153 Mbps, while the median 5G bandwidth merely reaches 304 Mbps [27]. Therefore, plenty of works are dedicated to improving the initial window and many CDN providers have increased the initial window to 32 segments or even 100 segments [6, 23]. However, due to the wide range of bandwidths under mobile networks, it is difficult to obtain clear improvements for all path BDPs using static initial windows. There are also works that use dynamic initial window settings based on historical information, including coarse-grained user group information [21, 25] and fine-grained connection information [12, 13]. However, for mobile networks, initial window settings are not all-inclusive. Even if the historical information is accurate, the fluctuating mobile networks may invalidate the historical information. Therefore, WiseStart not only reuses the historical connection information, but also performs path probe and convergence accordingly.

In addition to the initial CWND, it is critical to properly design the timing of the exit from slow start. The CWND growth exponentially in the slow start phase. As the CWND approaches the BDP of the path, it may also overshoot the link capacity, causing unnecessary congestion, which was also observed in our experiments (Fig. 9). Therefore, some researches focused on the exit point of slow start, such as setting a new threshold to decelerate the slow start [4] and using richer metrics (e.g. RTT) to determine [9, 14]. WiseStart can be combined with any of these optimization methods, and in our experiments, WiseStart uses Hystart++ algorithm [9].

## 6 Conclusion

We propose WiseStart, a new slow start mechanism for mobile web services. WiseStart reuses priori knowledge for the new connection, continuously probes the new connection to handle the fluctuating network conditions, and carefully adapts to the possible application-limit scenarios. We implement WiseStart in a popular mobile web service, and evaluate it in both emulated and production environments. Experiments show that WiseStart reduces the First AFT by 16.15% to 52.34% in different scenarios.

This work does not raise any ethical issues.

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
