# OpenReview forum: "Cold Start or Hot Start? Robust Slow Start in Congestion Control with A Priori Knowledge for Mobile Web Services"
_ACM.org/TheWebConf/2024/Conference — TheWebConf24_

### Official Review · Reviewer_N7BZ · 2023-11-23

**Novelty:** 6
**Technical Quality:** 5

**Review:**

* The paper introduces WiseStart, a novel "hot-start-based" slow start mechanism that utilizes a priori knowledge to optimize the initial parameters of slow start in mobile web services​​.
* Extensive experiments were conducted in a production environment of a popular mobile web service with over 10 million daily active users, demonstrating WiseStart's effectiveness​​.
* WiseStart's design continuously probes new connections to adapt to fluctuating network conditions and unpredictable application traffic, enhancing its robustness​​.
* WiseStart includes mechanisms to adapt to application-limit scenarios, contributing significantly to the reduction of First AFT​​.

Cons:
*  Initially, WiseStart had a hit rate of only 24.59% in the first hour, gradually increasing over time. This suggests initial complexity in its effectiveness and a learning curve for optimal performance​​.
* While effective for mobile web services, the applicability of WiseStart to other types of services or broader network conditions is not explicitly discussed.
* The paper could expand on how WiseStart integrates or complements other emerging network technologies and protocols, especially in increasingly diverse network environments​​.

**Questions:**

* Could you elaborate on how WiseStart performs in scenarios where historical data is limited or unreliable?
* How does WiseStart integrate with or complement other emerging network technologies and protocols?
* How scalable is WiseStart in terms of handling a growing number of users, and what is its long-term performance stability?

**Reviewer Confidence:**

3: The reviewer is confident but not certain that the evaluation is correct

**Scope:**

4: The work is relevant to the Web and to the track, and is of broad interest to the community

---

### Official Review · Reviewer_iVGY · 2023-11-23

**Novelty:** 4
**Technical Quality:** 4

**Review:**

This paper presents an innovative approach to the longstanding slow start issue in TCP or QUIC protocols by introducing a strategy termed "hot start," which utilizes an aggressive congestion window under optimal network conditions. The authors propose a novel combination of historical network performance data, called prior knowledge, and a robust measurement method capable of addressing various challenges, including rate limiting. The real-world application of the WiseStart algorithm demonstrates its effectiveness and fairness.

Pros:

- The paper tackles a critical issue, offering a sensible and theoretically sound solution.

- The evaluation is thorough, with the algorithm achieving commendable performance results.

Cons:

- The manuscript could benefit from improved clarity and detail. For instance,

A. The acronym 'FCT' needs to be defined upon its first use for the reader's understanding.

B. The document contains multiple different descriptions of 'BtlBW' and should be consistent.

C. Reference numbers should not be placed within numerical values (e.g., between '4' and '20') to avoid confusion.

D. The legend for Figure 9 should not solely depend on color coding, as this can be problematic for readers with color vision deficiencies or black-and-white printouts.

E. Regarding Algorithm 1, there is a concern when the congestion window (CWND) is smaller than 20*MSS and the ''Inflight'' data exceeds CWND, as the pacing_rate is adjusted twice, at lines 15 and 12. Clarification is needed to confirm if this is a deliberate aspect of the algorithm.

F. Enhancing the pseudocode of Algorithm 1 to explicitly include the maintenance and decision phases would greatly aid in the reader's comprehension of the algorithm's structure and flow.

**Questions:**

- The paper posits that "current routing policies generally ensure that connections with the same source and destination IPs will experience similar path conditions." However, our recent traceroute measurements to the same destination IP from an identical server show that 28% of probes followed different paths. What impact might this have for the algorithm's performance?

- What is the impact of the additional probe latency introduced by the algorithm compared to the method without measurement? Is there a scenario in which the algorithm is always in probing mode (can not reach the capacity of the path), and if so, what are the consequences?

- In Figure 9, the algorithm exhibits a higher byte loss than S-Cubic (the blue one) when the bandwidth multiplier is set to 0.8. Could the authors explain this observation?

**Reviewer Confidence:**

3: The reviewer is confident but not certain that the evaluation is correct

**Scope:**

4: The work is relevant to the Web and to the track, and is of broad interest to the community

---

### Official Review · Reviewer_NbJ1 · 2023-11-23

**Novelty:** 5
**Technical Quality:** 6

**Review:**

Thank you for your submission. This work presents WiseStart, a mechanism for improving connection slow start performance. The key insights are in bootstrapping the performance using historic data, then implements a probing mechanism which attempts to adapt to changing conditions, either backing off or filling the newly available bandwidth accordingly.

The design of the system is motivated and communicated clearly. The evaluations are convincing, but some of the presentation is difficult to follow. In particular many of the plots are quite dense and warrant some further explanation and detail, as they don't obviously match the text.

One particularly clever component of the system attempts to account for mis-estimated bandwidth caused by limitations in application behavior. Similar accounting was performed in [1] — although in a slightly different setting. How do the methods here compare? Another challenge discussed in related work can be found in [2], where improvements to initial connection behavior were often limited by clients initial receive windows is that still a concern here? Has this changed dramatically in the QUIC setting?

Matching client IPs is a relatively narrow condition, requiring measurements from the same address (which, given ISP allocation schemes, CGNATs, etc., may not be the same user). Could this be expanded to a larger condition that largely describes the same networking conditions, e.g. a /24 or other scope that would allow for a better hit rate on the historical data.

Pro
- Clearly motivates a scenario in which slow start is hurting performance
- Develops a system that addresses these issues
- Demonstrates the effectiveness of the system in number of settings and a robust set of baselines
- Focuses on user performance metrics like AFT, while showing improvement in underlying measurements such as RCT

Con:
- Some results communicated in unclear ways (presentation issues rather than
fundamental issues)

Detailed Comments:

- 2.2 - How do you measure maximum available bandwidth? As reported by the congestion window or some kind of measured value?

- Why only look at an individual client IP? Why not a wider set of IPs, e.g /24 or other subsets of traffic that are likely to have similar performance characteristics, but could dramatically reduce the storage overhead

- Is this strictly limited to mobile access? Are we considering access technology? e.g. just handsets? Cellular or including wifi?

-If we are focused on cellular are we concerned about the impact cellular packet gateways maybe have on the server side view of the interaction?

- 4.1 - I like the set of comparison baselines

- Does this have impacts on downstream dynamics of the system, eg. rct beyond the 1s mark

- 4.2 The numbers reported are confusing — the RCT is reduced by x% — is that to say the median is reduced? The text reports a tail latency decrease of 52.34%, but the tail  appears at about .6, higher than cubic and cubic 64? However the normalized latency in 6b shows lower values for wisestart for all shown percentiles. Greater clarity on which numbers are reporting what % to which baseline would strengthen this section significantly

- 4.2 - These hit rates seem very low! See note above regarding larger aggregates.

- One challenge observed in previous studies [2] is a limited receive window on clients, restricting the amount of data client will accept in the first place and artificially capping the benefits of the system. Is that still a concern in the modern QUIC setting?

- 4.3 - Numbers reported are again difficult to follow, but the picture is more clear here: wisestart AFT sits lower than others

- How is the s-cubic degradation shown in this experiment? Is there a way that can be read from this figure?

- Figure 9A is again difficult to read and follow. The text doesn't seem to address that wisestart has the highest (non-cubic) packet loss at bw multipliers near 1? This is much better described in the bw increasing case.

- 4.4.2 - greatly appreciate a component analysis!

[1] FaceBooks Internet Performance from Facebook's Edge, B, Schlinker, et al, IMC 2019

[2] Riptide: Jump-Starting Back-Office Connections in Cloud Systems, M. Flores, et al, IDCDS '16

**Questions:**

Could this system be improved by considering a larger unit of client history? For example aggregating client behaviors on /24, or similar units which are likely to experience similar behavior? Would the historical behavior be improved it were aged out?

**Reviewer Confidence:**

4: The reviewer is certain that the evaluation is correct and very familiar with the relevant literature

**Scope:**

4: The work is relevant to the Web and to the track, and is of broad interest to the community

---

### Official Review · Reviewer_9ABD · 2023-11-24

**Novelty:** 5
**Technical Quality:** 6

**Review:**

## Summary

A common metric of web performance, especially for mobile applications, is the above-the-fold time of the first page. Given typical page sizes, this is likely to be downloaded during the slow-start phase of the connection. However, slow-start, as widely deployed, is a “cold start” mechanism, using hard-coded parameters that don’t necessarily match with conditions or application traffic. The paper proposes WiseStart, which is a “hot start” based mechanism, using a priori knowledge to initialise the slow-start phase, before continuously probing to meet network conditions. The paper presents experiments that show a reduction in first AFT by about 16%.

## Reasons to accept

- The paper is mostly well written, and presents an interesting solution, that is well motivated and evaluated.
- The results contribute to the state-of-the-art.

## Reasons to reject

- It isn't clear how generalisable the results are of this work: even for the large mobile application platform that is used in the paper, the hit rate (i.e., the number of users for whom a priori information is available) is about 50% after 60 hours. That's likely to be much lower for less popular applications.

**Questions:**

- The related work section mentions that WiseStart uses HyStart++ -- do the other algorithms that WiseStart is evaluated against also use this algorithm?

**Reviewer Confidence:**

2: The reviewer is willing to defend the evaluation, but it is likely that the reviewer did not understand parts of the paper

**Scope:**

3: The work is somewhat relevant to the Web and to the track, and is of narrow interest to a sub-community

---

### Official Review · Reviewer_idGZ · 2023-11-25

**Novelty:** 4
**Technical Quality:** 4

**Review:**

This paper presents an innovative solution addressing a critical issue in mobile web services, supported by empirical evidence. However, its generalizability, evaluation, long-term performance, and consideration of external factors could be further explored for a more robust assessment.

**Strengths**:

Significance: The paper addresses a significant issue in mobile web services—optimizing the loading time of the first page, crucial for user experience. It recognizes the limitations of the traditional "cold start" mechanism in this context.

Novelty: The proposal of a new mechanism, WiseStart, introduces a "hot-start-based" approach that leverages a priori knowledge and adapts dynamically to network conditions and application scenarios. This innovation suggests a promising solution to the identified problem.

Real-world Implementation: Implementing WiseStart in a popular online mobile web service adds practical value. Conducting comprehensive experiments in a production environment lends credibility to the proposed solution's effectiveness.

**Weaknesses**:

Generalizability: While the implementation and experiments were conducted on a popular mobile web service, the paper may lack generalizability. The effectiveness of WiseStart across various types of web services or under different conditions remains unclear.

Evaluation: While the paper quantifies the improvements in AFT and RCT, it might benefit from including a broader set of metrics to evaluate WiseStart's overall impact on user experience or other performance aspects beyond connection establishment.

Validation of Long-term Performance: The paper demonstrates immediate improvements, but it could be strengthened by evaluating the long-term stability and performance of WiseStart over extended periods or under varying user loads.

**Questions:**

The experiments may not account for all potential external factors that could affect real-world performance. To enhance the robustness of the proposed solution, would the authors consider a wider range of scenarios and potential disruptions?

**Reviewer Confidence:**

1: The reviewer's evaluation is an educated guess

**Scope:**

2: The connection to the Web is incidental, e.g., use of Web data or API

---

### Decision · Program_Chairs · 2024-01-22

**Decision:**

Accept

**Comment:**

Reviewers agree that the problem of "cold start" is an important one and the WiseStart algorithm seems like a reasonable proposal. It also seems to have been deployed in-the-wild. However, numerous presentation as well as clarity issues about evaluation were discussed in the discussion phase. This should be incorporated into the next version. The focus is on congestion control for end user devices, which, while relevant to the Web community, may be of somewhat narrow interest.